# When Privacy Meets Partial Information: A Refined Analysis of Differentially Private Bandits

**Achraf Azize and Debabrota Basu**
Équipe Scool, Univ. Lille, Inria,
CNRS, Centrale Lille, UMR 9189- CRIStAL
F-59000 Lille, France
{achraf.azize,debabrota.basu}@inria.fr

## Abstract

We study the problem of multi-armed bandits with $\epsilon$-global Differential Privacy (DP). First, we prove *the minimax and problem-dependent regret lower bounds* for stochastic and linear bandits that quantify the hardness of bandits with $\epsilon$-global DP. These bounds suggest the existence of two hardness regimes depending on the privacy budget $\epsilon$. In the high-privacy regime (small $\epsilon$), the hardness depends on a coupled effect of privacy and partial information about the reward distributions. In the low-privacy regime (large $\epsilon$), bandits with $\epsilon$-global DP are not harder than the bandits without privacy. For stochastic bandits, *we further propose a generic framework to design a near-optimal $\epsilon$-global DP extension of an index-based optimistic bandit algorithm.* The framework consists of three ingredients: the Laplace mechanism, arm-dependent adaptive episodes, and usage of only the rewards collected in the last episode for computing private statistics. Specifically, we instantiate $\epsilon$-global DP extensions of UCB and KL-UCB algorithms, namely AdaP-UCB and AdaP-KLUCB. AdaP-KLUCB *is the first algorithm that both satisfies $\epsilon$-global DP and yields a regret upper bound that matches the problem-dependent lower bound up to multiplicative constants.*

## 1 Introduction

Multi-armed bandit problems, in short *bandits*, are a model for sequential decision-making with partial information (Lattimore and Szepesvári, 2018). In bandits, a learner sequentially interacts with an environment, which is a set of unknown distributions (or arms or actions), over $T \in \mathbb{N}$ steps. $T$ is referred to as the horizon. At each step $t \in \{1, \dots, T\}$, the learner chooses an arm $A_t$ from $\{1, \dots, K\}$ and the environment reveals a reward $r_t$ from the distribution $\nu_{A_t}$. The learner's objective is to maximise its cumulative reward $\sum_{t=1}^{T} r_t$. An equivalent performance metric for a bandit algorithm $\pi$ is *regret*. Regret is the difference between the expected cumulative reward collected by pulling the optimal arm $a^*$ for $T$ times and the expected cumulative reward obtained by using $\pi$. Regret is the price paid by the bandit algorithm due to partial information about the reward distributions. *The goal of a bandit algorithm is to minimise its regret.* Bandits, introduced in (Thompson, 1933) for medical trials, are widely studied and deployed in real-life applications, such as online advertising (Schwartz et al., 2017), recommendation systems (Silva et al., 2022) and investment portfolio design (Silva et al., 2022). Sensitive data of individuals, such as health conditions, personal preferences, financial status etc., are utilised in these applications, which raises the concern about privacy.

**Example 1.** *In bandit-based recommendation systems, the learning algorithm selects an item for each user. The user decides whether or not to click the recommendation. Based on the click feedback, the bandit algorithm improves its subsequent recommendations. Here, an action corresponds to an*

36th Conference on Neural Information Processing Systems (NeurIPS 2022).

*item and a reward corresponds to whether a user clicks on the item (1) or not (0). A change in a user's preference prompts changes in the algorithm's output. As a result, even if the click feedback is kept private, the user's private information, i.e. their preference over items, is revealed. Privacy protection aims to prevent the algorithm's output from revealing the preferences of any specific user.*

This example demonstrates the need for privacy in bandits. In this paper, we use Differential Privacy (DP) (Dwork and Roth, 2014) as the framework for privacy. DP ensures that an algorithm's output is unaffected by changes in input at a single data point. By limiting the amount of sensitive information that an adversary can deduce from the output, DP renders an individual corresponding to a data point 'indistinguishable'. A calibrated amount of noise is injected into an algorithm to ensure DP. The noise scale is set to be proportional to the algorithm's sensitivity and inversely proportional to the privacy budget $\epsilon$.

To address the privacy issues in bandit applications, the problem of Differentially Private Bandits is coined and studied under different settings, such as stochastic bandits (Mishra and Thakurta, 2015; Tossou and Dimitrakakis, 2016; Sajed and Sheffet, 2019; Hu et al., 2021), adversarial bandits (Tossou and Dimitrakakis, 2017), and linear contextual bandits (Shariff and Sheffet, 2018; Neel and Roth, 2018). Also, multiple adaptations of DP, namely local and global, are proposed for bandits (Basu et al., 2019). *Local DP* aims to preserve the privacy of the sequence of rewards obtained by sending noisy rewards to the algorithm (Duchi et al., 2013). *Global DP* allows the algorithm to access rewards without noise and aims to keep the sequence of rewards private while only the sequence of actions taken by the algorithm is produced to the public (Basu et al., 2019). Though local DP provides stronger privacy as the data curator has no access to the original reward stream, it injects too much noise that leads to higher regret. Also, the fundamental hardness of local DP in bandits in terms of regret lower bound and also corresponding optimal algorithms are well-understood (Zheng et al., 2020). Thus, in this paper, *we focus on the bandit problems with $\epsilon$-global DP.* We aim to address two questions:

1. *What is the fundamental hardness of differentially private bandits with global DP expressed in terms of the regret lower bound?*
2. *How to design an algorithmic framework that converts an optimistic and near-optimal bandit algorithm into a near-optimal bandit algorithm satisfying global DP?*

**Our Contributions.** These questions have led to the following contributions:

1. *Hardness as Lower Bounds:* We derive both the minimax (worst-case) and problem-dependent lower bounds on the regret of bandits with $\epsilon$-global DP for stochastic bandits (Thm. 2 and 3, Sec. 3). Both the bounds show that the hardness depends on a trade-off between the privacy budget $\epsilon$ and the distinguishability gap of a bandit environment. If the $\epsilon$ is bigger than the distinguishability gap, a bandit with global DP is not harder than a non-private bandit problem. Additionally, our problem-dependent regret lower bound (Thm. 3) provides a novel observation that the difficulty of a bandit problem with global DP depends on the TV-indistinguishability gap ($t_{inf}$, Thm. 3). This was not known in the regret lower bound in (Shariff and Sheffet, 2018), where the privacy dependent term in regret, i.e. $\frac{K \log(T)}{\epsilon}$ is independent of the hardness of the bandit instance. Our lower bounds explicates this missing link between the interaction of privacy and partial information. We also extend our techniques to derive the lower bounds for linear bandits with a finite number of arms (Thm. 4 and 5). These lower bounds also reflect the same transition of hardness depending on $\epsilon$ and distinguishability gaps.

2. *Algorithm Design:* Optimistic bandit algorithms used the empirical mean and variance of observed rewards to compute indexes and use them to select an action. We propose three fundamental strategies to design a private bandit algorithm from an optimistic algorithm (Sec. 4)– a) add Laplacian noise to the empirical mean of each arm calibrated by the corresponding sensitivity, b) use adaptive episodes to compute the private empirical mean less number of times, and c) use only the observed rewards of the arm's *last active episode*[1] and forget everything before to keep the sensitivity of the empirical mean low. We deploy these techniques with UCB (Auer et al., 2002) and KL-UCB (Garivier and Cappé, 2011) algorithms for non-private bandits to propose two near-optimal and $\epsilon$-global DP bandit algorithms, AdaP-UCB and AdaP-KLUCB (Sec. 4). Both of these algorithms achieve near-optimal problem-dependent regret and also reflect a transition in hardness from a low to a high privacy regime

---

[1]The last active episode of an arm is the last episode in which that arm was played.

(Thm. 7 and 8). Both theoretical (Table 1) and experimental (Sec. 5) results demonstrate optimality of AdaP-KLUCB than existing algorithms.

3. *Technical Tools:* (a) To derive the lower bound, we extend the Karwa-Vadhan lemma (Lemma 6.1, (Karwa and Vadhan, 2017)) to the sequential setting (Lemma 2). (b) We present a novel sequential information processing lemma under $\epsilon$-global DP (Thm. 10) that controls the difference between the outcome streams of a differentially private policy when interacting with two different bandit instances. (c) This leads to a generic proof structure, utilised to generate refined regret lower bounds under $\epsilon$-global DP for different settings. (d) To derive the regret upper bound, we develop a novel and general analysis of optimistic algorithms with adaptive episodes.

Our regret lower and upper bounds close the open question posed in (Tenenbaum et al., 2021), i.e. the problem-dependent regret bounds of differentially private bandits should depend on the KL-divergence between the reward distributions.

**Related Works: Lower Bound.** (Shariff and Sheffet, 2018) first proposed a problem-dependent lower bound on regret, $\Omega(\max\{\sum_{a \neq a^*} \frac{\log T}{\Delta_a}, \frac{K \log(T)}{\epsilon}\})^2$, for stochastic bandits with $\epsilon$-global DP. But this bound is restricted to Bernoulli distributions of reward. In this paper, *we provide the first problem-dependent regret lower bound that is valid for any reward distribution. This lower bound shows that the effect of privacy does not only depend on $\epsilon$ but also on the total variation distance corresponding to the environment.* Thus, *it explicates the coupled effect of privacy and partial information in a high-privacy regime*, which was not observed before. To derive this tighter lower bound, we extend the Karwa-Vadhan lemma (Lemma 6.1, (Karwa and Vadhan, 2017)) to a sequential setting and also propose a generic proof structure leading to *problem-dependent and minimax lower bounds for stochastic and linear bandits*. To our knowledge, *these are the first regret lower bounds for linear bandits with $\epsilon$-global DP*. Basu et al. (2019) also proposed a minimax regret lower bound for stochastic bandits with $\epsilon$-global DP. However, our minimax lower bound (Thm. 2) is tighter and does not need to assume that the reward distributions are Lipschitz continuous.

**Related Works: Bandit Algorithms with $\epsilon$-global DP.** DP-UCB (Mishra and Thakurta, 2015; Tossou and Dimitrakakis, 2016) was the first global DP version of UCB. DP-UCB uses the tree-based mechanism (Dwork et al., 2010a; Chan et al., 2011) to compute the sum of rewards. The tree mechanism maintains a binary tree of depth $\log(T)$ over the $T$ streaming observations, where each node in this tree holds an i.i.d sample from a Laplace distribution with zero mean and scale $(\log(T)/\epsilon)$. At each step $t$, the mechanism yields the sum of the first $t$ observations and the $\log(T)$ nodes on the root-to-the-leaf path in the binary tree as the private empirical mean. As a result, the noise added to the UCB index per time-step is $\mathcal{O}\left(\log(T)^{2.5}/\epsilon\right)$, which is responsible for the extra multiplicative factor $\log(T)^{1.5}$ in regret compared to the lower bound. DP-SE (Sajed and Sheffet, 2019) was the first $\epsilon$-global DP algorithm to eliminate the additional multiplicative factor $\log(T)^{1.5}$. DP-SE is an $\epsilon$-global DP version of the Successive Elimination algorithm (Even-Dar et al., 2002). However, the drawbacks were that the algorithm was not anytime, and was optimal only asymptotically, i.e. the horizon should be big enough. On the other hand, a careful analysis of the algorithm suggests that what made the algorithm optimal was the fact that the algorithm was run in doubling episodes, where the private means computed at the end of the episode to decide which arms to eliminate, were only computed using the samples collected from that episode. *We detect these ingredients and generalise them to propose a general framework to make any optimist index-based policy optimal followed by upper bounds matching the lower bounds (Table 1).* Similar techniques have been deployed to design a global DP extension of UCB in (Hu et al., 2021) but they have an additive factor $\log(T)$ extra in regret than AdaP-UCB and do not match our proposed regret lower bound. Also, DP-TS (Hu and Hegde, 2022) aims to achieve $\epsilon$-global DP with a Thompson sampling based approach. The "lazy" version of DP-TS achieves a similar regret as AdaP-UCB, but does not achieve the refined lower bound of Thm. 3 with the indistinguishability gaps. Even in non-private bandits, Thompson Sampling (Agrawal and Goyal, 2013) is not known to achieve the Kullback-Leibler (KL) indistinguishability gap while KL-UCB (Garivier and Cappé, 2011) is known to achieve it. *Our work leads to the first near-optimal bandit algorithm with $\epsilon$-global DP, namely* AdaP-KLUCB.

---

[2] Here, $a^*$ is the optimal arm with mean reward $\mu^*$ and $\Delta_a \triangleq \mu^* - \mu_a$ is the suboptimality gap of arm $a$.

[3] The number of private empirical mean reward statistics computed by the algorithm.

[4] Not using the full history of rewards to compute the private empirical mean reward statistics.

Table 1: A comparison of $\epsilon$-global DP algorithms for bandits.

| Algorithm | Regret | # Private Means [3] | Anytime | Forgetfulness [4] |
|---|---|---|---|---|
| DP-UCB (Mishra and Thakurta (2015); Tossou and Dimitrakakis (2016)) | $\mathcal{O}\left(\frac{\log(T)^{2.5}}{\epsilon} + \sum_{a \neq a^*} \frac{\log(T)}{\Delta_a}\right)$ | $KT$ | Yes | No |
| DP-SE (Sajed and Sheffet, 2019) | $\mathcal{O}\left(\frac{K\log(T)}{\epsilon} + \sum_{a \neq a^*} \frac{\log(T)}{\Delta_a}\right)$ | $\mathcal{O}(\log(T))$ | No | Yes |
| AdaP-UCB | $\mathcal{O}\left(\sum_{a \neq a^*} \frac{\Delta_a \log(T)}{\min(\Delta_a^2, \epsilon \Delta_a)}\right)$ | $\mathcal{O}(\log(T))$ | Yes | Yes |
| AdaP-KLUCB | $\mathcal{O}\left(\sum_{a \neq a^*} \frac{\Delta_a \log(T)}{\min(d(\mu_a, \mu^*), \epsilon \Delta_a)}\right)$ | $\mathcal{O}(\log(T))$ | Yes | Yes |

## 2 Background: Differential Privacy and Bandits

**Differential Privacy (DP).** DP renders an individual corresponding to a datum indistinguishable by constraining the output of an algorithm to be almost the same under a change in one input datum.

**Definition 1** $((\epsilon, \delta)$-DP (Dwork and Roth, 2014)). *A randomised algorithm $\mathcal{A}$ satisfies $(\epsilon, \delta)$-Differential Privacy (DP) if for any two neighbouring datasets[5] $\mathcal{D}$ and $\mathcal{D}'$, and for all sets of output $\mathcal{O} \subseteq \mathrm{Range}(\mathcal{A})$*

$$\Pr[\mathcal{A}(\mathcal{D}) \in \mathcal{O}] \leq e^\epsilon \Pr[\mathcal{A}(\mathcal{D}') \in \mathcal{O}] + \delta, \tag{1}$$

*where the probability space is over the coin flips of the mechanism $\mathcal{A}$ and for some $(\epsilon, \delta) \in \mathbb{R}^{\geq 0} \times \mathbb{R}^{\geq 0}$. If $\delta = 0$, we say that $\mathcal{A}$ satisfies $\epsilon$-differential privacy.*

The Laplace mechanism (Dwork et al., 2010b; Dwork and Roth, 2014) ensures $\epsilon$-DP by injecting a random noise to the output of the algorithm that is sampled from a calibrated Laplace distribution (as specified in Theorem 1). We use $Lap(b)$ to denote the Laplace distribution with mean 0 and variance $2b^2$.

**Theorem 1** ($\epsilon$-DP of Laplace Mechanism (Theorem 3.6, Dwork and Roth (2014))). *Let us consider an algorithm $f: \mathcal{X} \to \mathbb{R}^d$ with sensitivity $s(f) \triangleq \max_{\mathcal{D}, \mathcal{D}' \ s.t \ \|\mathcal{D} - \mathcal{D}'\|_{\mathrm{Hamming}} = 1} \|f(\mathcal{D}) - f(\mathcal{D}')\|_1$. Here, $\|\cdot\|_1$ is the $L_1$ norm on $\mathbb{R}^d$. If $d$ noise samples $\{N_i\}_{i=1}^d$ are generated independently from $Lap(\frac{s(f)}{\epsilon})$, the output injected with the noise, i.e. $f(\mathcal{D}) + [N_1, \ldots, N_d]$ satisfies $\epsilon$-DP.*

The Laplace mechanism is originally proposed to ensure DP when the input database is static. In a sequential setting like bandits, a mechanism must update the published statistics as new data items arrive, and thus DP definitions are extended accordingly (Basu et al., 2019).

**Stochastic Bandits with Global DP.** Now, we describe the general canonical bandit model (Section 4.6, (Lattimore and Szepesvári, 2018)) and define global Differential Privacy (global DP) in this setting. A bandit algorithm (or policy) interacts with an environment $\nu$ consisting of $K$ arms with reward distributions $\{\nu_a\}_{a=1}^K$ for a given horizon $T$ and produces a history $\mathcal{H}_T \triangleq \{(A_t, R_t)\}_{t=1}^T$. At each step $t$, the choice of the arm depends on the previous history $\mathcal{H}_{t-1}$. The reward $R_t$ is sampled from the reward distribution $\nu_{A_t}$ and is conditionally independent of the previous history $\mathcal{H}_{t-1}$. A bandit algorithm (or policy) $\pi$ can be represented by a sequence $(\pi_t)_{t=1}^T$, where $\pi_t : \mathcal{H}_{t-1} \to [K]$ is a probability kernel. Thus, we denote the probability of occurrence of a sequence of actions $a^T \triangleq [a_1, \ldots, a_T]$ given a sequence of rewards $r^T \triangleq [r_1, \ldots, r_T]$ as $\pi(a^T \mid r^T) \triangleq \prod_{t=1}^T \pi_t(a_t \mid a_1, r_1, \ldots, a_{t-1}, r_{t-1})$. *If we consider $a^T$ as the output of a bandit algorithm and $r^T$ as the input, $\pi(a^T \mid r^T)$ defines the corresponding probability distribution on the output space required for defining DP* (Eq. (1)). In order to define DP in the sequential setup of bandits, the event-level privacy under continuous observations (Dwork et al., 2010b) framework is adopted. In this framework, if two data sequences differ on a single entry at a single time-step $t$ and are identical on all other time-steps, they are called neighbouring data sequences. A sequential algorithm is differentially private if its output sequence is not distinguishable from two neighbouring input sequences. This framework leads to the *global DP* definition for bandits (Mishra and Thakurta, 2015; Basu et al., 2019).

**Definition 2** (Global DP for Bandits). *A bandit algorithm $\pi \in \Pi^\epsilon$ satisfies $\epsilon$-global DP, if $\pi(a^T \mid r^T) \leq e^\epsilon \pi(a^T \mid r'^T)$ for every action sequence $a^T \in [K]^T$ and every two neighbouring reward sequences $r^T, r'^T \in \mathbb{R}^T$, which means that $\exists j \in [1, T]$ such that $r_j \neq r'_j$ and $\forall \, t \neq j \, r_t = r'_t$. The set of all bandit algorithms satisfying $\epsilon$-global DP, i.e. $\Pi^\epsilon$, is called the $\epsilon$-global DP policy class.*

---

[5]Neighbouring datasets differ only in one entry, i.e $\|\mathcal{D} - \mathcal{D}'\|_{\mathrm{Hamming}} = 1$.

Table 2: Regret lower bounds for bandits with $\epsilon$-global DP

| | Minimax | Problem Dependent |
|---|---|---|
| **Stochastic Multi-armed bandit** | $\max\left(\frac{1}{27}\sqrt{T(K-1)}, \frac{1}{131}\frac{K-1}{\epsilon}\right)$ | $\sum_{a:\Delta_a>0}\frac{\Delta_a \log(T)}{\min(d_a, 6\epsilon t_a)}$ |
| **Stoachastic Linear bandit** | $\max\left(\frac{\exp(-2)}{8}d\sqrt{T}, \frac{\exp(-6)}{4}\frac{d}{\epsilon}\right)$ | $\inf_{\alpha\in[0,\infty)^{\mathcal{A}}}\sum_{a\in\mathcal{A}}\alpha(a)\Delta_a \log(T)$ s.t. $\|a\|^2_{H_\alpha^{-1}} \leq \Delta_a \min\left(\frac{\Delta_a}{2}, 3\epsilon\rho(\mathcal{A})\right)$ |

Now, we formally define the regret of a bandit algorithm with $\epsilon$-global DP, namely $\pi^\epsilon$, as

$$\text{Reg}_T(\pi^\epsilon, \nu) \triangleq T\mu^*(\nu) - \mathbb{E}_{\nu\pi^\epsilon}\left[\sum_{t=1}^T R_t\right] = \sum_{a=1}^K \Delta_a \mathbb{E}_{\nu\pi^\epsilon}\left[N_a(T)\right]. \quad (2)$$

$N_a(T) \triangleq \sum_{t=1}^T \mathbb{1}\{A_t = a\}$ is the number of times the arm $a$ is played till $T$. $\Delta_a \triangleq \mu^*(\nu) - \mu_a$ is the suboptimality gap of the arm $a$. The expectation is taken with respect to the probability measure on action-reward sequences induced by the interaction of $\pi^\epsilon$ and $\nu$. *The objective of the bandit algorithm $\pi^\epsilon$ is to satisfy $\epsilon$-global DP while minimising the regret over horizon $T$. A regret lower bound of any $\epsilon$-global DP bandit algorithm in $\Pi^\epsilon$ quantifies the fundamental hardness of this problem.*

# 3 Regret Lower Bounds for Bandits with $\epsilon$-global DP

The first question is how much additional regret we have to endure in bandits with $\epsilon$-global DP than the bandits without privacy. The lower bound on regret provides insight into the intrinsic hardness of the problem and serves as a target for the optimal algorithm design. In this section, *we prove minimax and problem-dependent lower bounds for stochastic and linear bandits under $\epsilon$-global DP.* The proposed lower bounds are summarised in Table 2. We defer the proof details to the supplementary.

**Stochastic Bandits.** First, we consider the stochastic bandit problem with $K$-arms (as in Section 2).

*Minimax Regret.* We derive the minimax regret lower bound in this bandit setting with $\epsilon$-global DP. Minimax regret is the lowest achievable regret by any bandit algorithm under the worst environment among a family of environments $\mathcal{E}^K$ with $K$ arms under consideration.

$$\text{Reg}_{T,\epsilon}^{\text{minimax}} = \inf_{\pi^\epsilon\in\Pi^\epsilon}\sup_{\nu\in\mathcal{E}^K}\text{Reg}_T(\pi^\epsilon, \nu)$$

**Theorem 2** (Minimax lower bound). *For any $K > 1$ and $T \geq K - 1$, and $\epsilon > 0$, the minimax regret of stochastic bandits with $\epsilon$-global DP satisfies*

$$\text{Reg}_{T,\epsilon}^{minimax} \geq \max\left\{\frac{1}{27}\underbrace{\sqrt{T(K-1)}}_{without\ global\ DP}, \quad \frac{1}{131}\underbrace{\frac{K-1}{\epsilon}}_{with\ \epsilon\text{-}global\ DP}\right\}. \quad (3)$$

*Consequences of Theorem 2.* Equation (3) shows two regimes of hardness: *high-privacy*, corresponding to lower values of $\epsilon$, and *low-privacy*, corresponding to higher values of $\epsilon$. In the high-privacy regime, specifically for $\epsilon < \frac{131}{27}\sqrt{\frac{(K-1)}{T}}$, the hardness depends only on the number of the arms $K$ and the privacy budget $\epsilon$ and is higher than the lower bound for bandits without privacy. In the low-privacy regime, i.e. for $\epsilon \geq \frac{131}{27}\sqrt{\frac{(K-1)}{T}}$, the lower bound coincides with that of the bandits without privacy. This indicates the phenomenon that for higher values of $\epsilon$, signifying lower privacy, bandits with global DP are not harder than the bandits without privacy. Especially, for significantly large $T$, the threshold between low and high privacy regimes is smaller than most of the practically used privacy budget values. For example, if $T = 10^6$ and $K = 100$, the bandits with and without global DP are equivalently hard for any privacy budget $\epsilon \geq 0.05$. This shows that for stochastic bandits, we can deploy very low privacy budgets ($\epsilon$) without losing anything in performance.

*Problem-dependent Regret:* Similar to the lower bounds specific to the Bernoulli reward distributions (Shariff and Sheffet, 2018), the minimax bound indicates a separation between hardness due to global DP and partial information. To understand whether this observation is an artefact of the

Bernoulli distribution and the worst-case environment considered in minimax bounds, we derive a lower bound on the problem-dependent regret (Equation (2)) for general reward distributions.

Before deriving the lower bound, we define two information-theoretic terms that quantify the distinguishability of the specific bandit environment $\nu$ where the algorithm is operating from all other bandit environments with $K$ arms and finite means. If $\mathcal{M}$ is a set of distributions with finite means, and $\mu(P)$ is the mean of the reward distribution $P \in \mathcal{M}$, we define the *KL-distinguishability gap* as $d_{\inf}(P, \mu^*, \mathcal{M}) \triangleq \inf_{P' \in \mathcal{M}} \{D_{\mathrm{KL}}(P \parallel P') : \mu(P') > \mu^*\}$ and the *TV-distinguishability gap* as $t_{\inf}(P, \mu^*, \mathcal{M}) \triangleq \inf_{P' \in \mathcal{M}} \{\mathrm{TV}(P \parallel P') : \mu(P') > \mu^*\}$. These two quantities indicate the minimum dissimilarity, in terms of KL-divergence and Total Variation (TV) distance, of the reward distribution of the optimal arm $a^*$ with any other distribution with a finite mean higher than $\mu^*$. For bandits without DP, the inverse of the KL-distinguishability gap quantifies the hardness to identify the optimal arm of the environment $\nu$ under partial information (Lai and Robbins, 1985).

**Theorem 3** (Problem-dependent Regret Lower Bound). *Let the environment $\mathcal{E}$ be a set of $K$ reward distributions with finite means and a policy $\pi^\epsilon \in \Pi_{cons}(\mathcal{E}) \cap \Pi^\epsilon$ be a consistent policy[6] over $\mathcal{E}$ satisfying $\epsilon$-global DP . Then, for all $\nu = (P_i)_{i=1}^K \in \mathcal{E}$, it holds that*

$$\liminf_{T \to \infty} \frac{\mathrm{Reg}_T(\pi^\epsilon, \nu)}{\log(T)} \geq \sum_{a : \Delta_a > 0} \frac{\Delta_a}{\min\left(\underbrace{d_{\inf}(P_a, \mu^*, \mathcal{M}_a)}_{\text{without global DP}}, \underbrace{6\,\epsilon\,t_{\inf}(P_a, \mu^*, \mathcal{M}_a)}_{\text{with } \epsilon\text{-global DP}}\right)}. \tag{4}$$

*Consequences of Theorem 3.* Now, we summarise the interesting observations led by the lower bound.

1. *Universality:* The lower bound of Theorem 3 holds for any environment with $K$ arms and reward distributions with finite means. This is the first general lower bound for bandits with $\epsilon$-global DP.

2. *For Bernoulli distributions of reward:* TV-distinguishability gap $t_{\inf}(P_a, \mu^*, \mathcal{M}_a) = \Delta_a$, i.e. the suboptimality gap of arm $a$, and the KL-distinguishability gap $d_{\inf}(P_a, \mu^*, \mathcal{M}_a) \geq 2\Delta_a^2$. Thus, our problem-dependent lower bound reduces to $\Omega(\sum_{a \neq a^*} \frac{\log T}{\min\{\Delta_a, \epsilon\}})$. For Bernoulli rewards, our lower bound is able to retrieve the lower bound of (Shariff and Sheffet, 2018) with explicit constants.

3. *High- and Low-privacy Regimes:* Like the minimax regret bound, the problem-dependent regret also indicates two clear regimes in regret due to high and low privacy (resp. small and large privacy budgets $\epsilon$). *In the low-privacy regime, i.e. for $\epsilon \geq \frac{d_{\inf}(P_a, \mu^*, \mathcal{M}_a)}{6 t_{\inf}(P_a, \mu^*, \mathcal{M}_a)}$, the regret achievable by bandits with global DP and without global DP are same.* Thus, there is no loss in performance due to privacy in this regime. *In the high-privacy regime, i.e. for $\epsilon < \frac{d_{\inf}(P_a, \mu^*, \mathcal{M}_a)}{6 t_{\inf}(P_a, \mu^*, \mathcal{M}_a)}$, the regret depends on a coupled effect of privacy and partial information.* This effect is quantified by the inverse of the privacy budget times the inverse of the TV-distinguishability gap. This interaction between privacy and partial information was not explicated by any of the existing lower bounds. In Section 4, we propose the AdaP-KLUCB algorithm that demonstrates the same interactive effect in its regret upper bound.

**Stochastic Linear Bandits.** In order to illustrate the generality of our results, and also to investigate how partial information and privacy in structured bandits, we derive minimax and problem-dependent regret lower bounds for stochastic linear bandits (Lattimore and Szepesvari, 2017). To perform the regret analysis, we consider a simple linear model with parameter $\theta \in \mathbb{R}^d$ and Gaussian noise. It implies that for an action $A_t \in \mathcal{A} \subseteq \mathbb{R}^d$ the reward is $R_t = \langle A_t, \theta \rangle + \eta_t$, where $\eta_t \sim \mathcal{N}(0, 1)$ is a sequence of independent Gaussian noises. The regret of a policy is $\mathrm{Reg}_T(\mathcal{A}, \theta) \triangleq \mathbb{E}_\theta\left[\sum_{t=1}^T \Delta_{A_t}\right]$, where suboptimality gap $\Delta_a \triangleq \max_{a' \in \mathcal{A}} \langle a' - a, \theta \rangle$, and $\mathbb{E}_\theta[\cdot]$ is the expectation with respect to the measure o outcomes induced by the interaction of the policy and the linear bandit determined by $\theta$. Given this structure, we state the minimax and problem-dependent regret lower bounds for stochastic linear bandits.

---

[6] A policy $\pi$ is called consistent over a class of bandits $\mathcal{E}$ if for all $\nu \in \mathcal{E}$ and $p > 0$, it holds that $\lim_{T \to \infty} \frac{R_T(\pi, \nu)}{T^p} = 0$. We denote the class of consistent policies over a set of environments $\mathcal{E}$ as $\Pi_{cons}(\mathcal{E})$.

**Theorem 4** (Minimax regret lower bound). *Let $\mathcal{A} = [-1, 1]^d$ and $\Theta = \mathbb{R}^d$. Then, for any $\epsilon$-global DP policy, we have that*

$$\text{Reg}_T^{minimax}(\mathcal{A}, \Theta) \geq \max\Big\{\underbrace{\frac{\exp(-2)}{8}d\sqrt{T}}_{\text{without global DP}}, \underbrace{\frac{\exp(-6)}{4}\frac{d}{\epsilon}}_{\text{with } \epsilon\text{-global DP}}\Big\}. \tag{5}$$

**Theorem 5** (Problem-dependent regret lower bound). *Let $\mathcal{A} \subset \mathbb{R}^d$ be a finite set spanning $\mathbb{R}^d$ and $\theta \in \mathbb{R}^d$ be such that there is a unique optimal action. Then, any consistent and $\epsilon$-global DP bandit algorithm satisfies*

$$\liminf_{T \to \infty} \frac{\text{Reg}_T(\mathcal{A}, \theta)}{\log(T)} \geq c_\epsilon(\mathcal{A}, \theta), \tag{6}$$

*where the* structural distinguishability gap *is the solution of a constraint optimisation*

$$c_\epsilon(\mathcal{A}, \theta) \triangleq \inf_{\alpha \in [0, \infty)^{\mathcal{A}}} \sum_{a \in \mathcal{A}} \alpha(a)\Delta_a, \text{ such that } \|a\|_{H_\alpha^{-1}}^2 \leq \min\Big\{\underbrace{0.5\Delta_a^2}_{\text{without global DP}}, \underbrace{3\epsilon\rho_a(\mathcal{A})\Delta_a}_{\text{with } \epsilon\text{-global DP}}\Big\}$$

*for all $a \in \mathcal{A}$ with $\Delta_a > 0$, $H_\alpha = \sum_{a \in \mathcal{A}} \alpha(a)aa^\top$, and a structure-dependent constant $\rho_a(\mathcal{A})$.*

*Remarks.* Theorems 4 and 5 are the first minimax and problem-dependent regret lower bounds for linear bandits. The minimax regret bound also shows a clear distinction between high- and low-privacy regimes for $\epsilon < 2\exp(-4)/\sqrt{T}$ and $\epsilon \geq 2\exp(-4)/\sqrt{T}$. For the problem-dependent bound, the difference between high- and low-privacy regimes is more subtle. The first constraint on $\|a\|_{H_\alpha^{-1}}^2$ is reminiscent of the Graves-Lai bound for structural bandits without privacy. The second constraint, i.e. $3\epsilon\rho_a(\mathcal{A})\Delta_a$, captures the interaction between privacy and partial information under linear structure and in the high-privacy regime.

**Proof Technique.** In order to prove the lower bounds, we adopt the general canonical bandit model. The high-level idea of proving bandit lower bounds is selecting two *hard* environments, which are hard to be statistically distinguished but are conflicting, i.e. actions that may be optimal in one is suboptimal in other. This is quantified by upper bounding the per-step KL-divergence between action distributions for such hard environments, and then plugging in this upper bound in the Bretagnolle-Huber inequality to obtain a regret lower bound. Though this proof technique works to quantify the hardness due to partial information, we need to upper bound the "confusion" created due to global DP. Existing proofs use the information-processing lemma of Karwa-Vadhan (Karwa and Vadhan, 2017). To prove a general bound, we derive a sequential version of this lemma. The sequential version leads to an upper bound of the KL-divergence dependent on both the total-variation distinguishability gap and the privacy budget, and enables us to show the coupled effect of privacy and partial-information in the high-privacy regimes. For details of the technical results, we refer to the Supplementary Material.

## 4 Stochastic Bandit Algorithms with $\epsilon$-global DP: AdaP-UCB & AdaP-KLUCB

In this section, *we present a framework (Algorithm 1) to convert index-based optimistic algorithms into algorithms satisfying $\epsilon$-global DP*. We instantiate this framework by proposing AdaP-UCB and AdaP-KLUCB, and derive corresponding regret upper bounds.

---
**Algorithm 1** A framework for $\epsilon$-global DP extension of an index-based bandit algorithm
---
1: **Input:** Privacy budget $\epsilon$, an environment $\nu$ with $K$ arms, parameter $\alpha > 3$
2: **Output:** A sequence of $T$-actions satisfying $\epsilon$-global DP
3: **Initialisation:** Choose each arm once and assign $t = K$
4: **for** $\ell = 1, 2, \ldots$ **do**                ▷ Adaptive episodes of reward subsequences
5:        Let $t_\ell = t + 1$
6:        Compute $A_\ell = \text{argmax}_a \, \text{I}_a^\epsilon(t_\ell - 1, \alpha)$    ▷ Arm selection with private indexes (Eqn. (7)-(8))
7:        Choose arm $A_\ell$ until round t such that $N_{A_\ell}(t) = 2N_{A_\ell}(t_\ell - 1)$       ▷ Doubling of episodes
8: **end for**
---

**Index-Based Algorithms and Private Empirical Mean.** We focus on index-based bandit algorithms that compute the empirical mean of rewards of each arm at each step. *Using the empirical means, they*

*compute an optimistic index for each arm that serves as a high-probability upper confidence bound on the true mean of the corresponding arm.* Examples of such algorithms with empirical mean-based indexes include UCB (Auer et al., 2002), MOSS (Audibert and Bubeck, 2010), KL-UCB (Garivier and Cappé, 2011), IMED (Honda and Takemura, 2015) etc. Here, the empirical mean is the main statistics of reward sequences used by the algorithms. Thus, by post-processing lemma, *designing an $\epsilon$-global DP bandit algorithm reduces to computing the empirical means privately.*

By Theorem 1, we know that adding a Laplacian noise with scale $s(\hat{\mu}_t)/\epsilon$ to each empirical mean turns it $\epsilon$-DP, where $s(\hat{\mu}_t)$ is the sensitivity of empirical mean $\hat{\mu}_t$ at step $t$. For $K$ arms and horizon $T$, an index-based bandit algorithm computes the empirical mean $KT$ times. Using the naïve composition of $\epsilon$-DP, a first baseline is to make each computed empirical mean $\frac{\epsilon}{KT}$-DP. This needs us to add at each step noise scaled linearly to $T$. This might lead the corresponding bandit algorithm to yield a linear regret. Thus, we need to add noise less number of times and with lower sensitivity.

**Empirical Mean using Reward Sub-sequence.** The improvement is invoked by the observation that a bandit algorithm does not need to calculate the empirical mean at each time-step using all the rewards observed till that step. Thus, motivated by the DP-SE (Sajed and Sheffet, 2019), we calculate the empirical mean less number of times by exploiting an episodic structure that we will explicit later. Specifically, we use only the rewards of the last episode. We formally express this trick in Lemma 1.

**Lemma 1.** *Let us define the private empirical mean of the rewards between steps $i$ and $j$ ($i < j$) as*

$$f^\epsilon\{r_i,\ldots,r_j\} \triangleq \frac{1}{j-i}\sum_{t=i}^{j} r_t + Lap\left(\frac{1}{(j-i)\epsilon}\right).$$

*If $1 < t_1 < \cdots < t_\ell < T$ and $r_t \in [0,1]$, the mechanism $g^\epsilon$ mapping the sequence of rewards $(r_1, r_2, \ldots, r_T)$ to $(\ell+1)$-private empirical means $(f^\epsilon\{r_1,\ldots,r_{t_1-1}\}, f^\epsilon\{r_{t_1},\ldots,r_{t_2-1}\}, \ldots, f^\epsilon\{r_{t_\ell},\ldots,r_T\})$ satisfies $\epsilon$-DP.*

Lemma 1 implies that, if we calculate the empirical mean of each arm $\ell + 1$ ($\ll T$) times on non-overlapping sub-sequences of the reward stream, we only need to ensure that each empirical mean is $\epsilon$-DP with respect to the corresponding reward sub-sequence. These $(\ell+1)$ private empirical means together ensure that the sequence of $(\ell+1)$ computed indexes and the resultant action sequence of length $T$ satisfy $\epsilon$-global DP by the post-processing lemma (Dwork and Roth, 2014). *Thus, we divide the horizon into $(\ell+1)$-episodes. This allows us to take $T$ actions only by computing $\ell + 1$ private empirical means with the rewards observed in the last active episode of the arm.* Specifically, for each $t \in [t_i, t_{i+1} - 1]$, we play the same arm that was decided at the beginning of the episode, i.e. at $t_i$, with the private empirical mean computed from the last active episode of that arm.

**Adaptive Episodes with Doubling.** In order to set the episode lengths, we focus on the specific structure of the bandit process. Specifically, we know that a sub-optimal arm $a$ should be sampled at least $n_a = \mathcal{O}\left(\frac{\log(T)}{\Delta_a^2}\right)$ times before discarding. Since the empirical means are computed using only rewards of one episode, to yield a near-optimal regret, the length of the episode should be greater at any given time than $n_a$. *To satisfy this criterion, we deploy adaptive episodes with doubling.*

**Example 2** (Illustration of Algortihm 1.)**.** *To clarify the schematic, we illustrate a few steps of executing Algorithm 1 in Figure 1. After playing each arm once, the first episode begins at $t_1$. To observe different ingredients, we focus on step $t_4 = 7$. The index of Arm 1 at $t_4$ uses the private empirical mean $\frac{r_4+r_5}{2} + Lap(\frac{1}{2\epsilon})$ to build a high probability upper bound of the real mean $\mu_1$ with confidence $t_4^{-\alpha}$. The index of Arm 2 uses $r_6 + Lap(\frac{1}{\epsilon})$. If we assume that the index of Arm 1 is higher at $t_4$, Arm 1 is played for a full episode from $t_4$ until $t_5 - 1$. The last time, when Arm 1 was played, the length of the episode was 2. Thus, following $t_4$, the length of the episode is doubled to 4.*

**Theorem 6** ($\epsilon$-global DP)**.** *If $I_a^\epsilon$ is computed using only the private empirical mean of the rewards collected in the last active episode of arm $a$, Algorithm 1 satisfies $\epsilon$-global DP.*

*Proof.* The main idea is that a change in reward will only affect the empirical mean calculated in one episode, which is made private using the Laplace Mechanism and Lemma 1. Since the actions are only calculated using the private empirical means, the algorithm is $\epsilon$-global DP following the post-processing lemma. We refer to Appendix D for a complete proof. □

Now, to concretise an algorithm, we only need to explicit how the indexes are calculated. In other words, we need to build a high-probability upper confidence bound on the true mean of an arm only using a private empirical mean. We instantiate the design details for AdaP-UCB and AdaP-KLUCB.

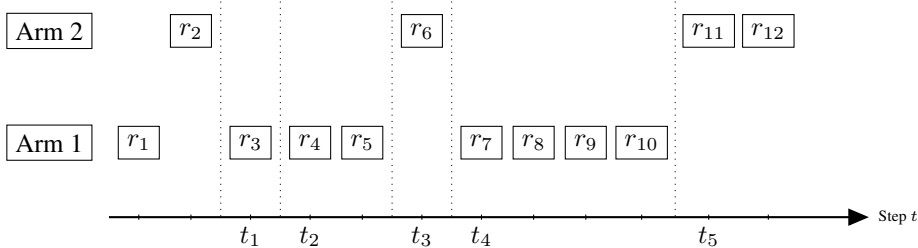

Figure 1: An illustration of adaptive episodes with per-arm doubling.

AdaP-UCB **and** AdaP-KLUCB. Let $\hat{\mu}_a^\ell$ be the empirical mean reward of arm $a$ computed using the samples collected between $t_{\psi_a(\ell)}$ and $t_{\psi_a(\ell)+1} - 1$. For an episode $\ell$, $\psi_a(\ell) = \ell_a$ is the last active episode of arm $a$. In Example 2, $\psi_1(4) = 2$ and $\psi_2(4) = 3$. Thus, due to the doubling of episode length, the empirical mean corresponds to $\frac{1}{2} N_a(t_\ell - 1)$ samples of arm $a$. Since the rewards are in $[0, 1]$, the private empirical mean as $\tilde{\mu}_{a,\epsilon}^\ell = \hat{\mu}_a^\ell + Lap\left(\frac{2}{\epsilon N_a(t_\ell - 1)}\right)$ satisfies $\epsilon$-DP (Theorem 1). Now, we want to ensure that $\mathrm{I}_a^\epsilon(t_\ell - 1, \alpha)$, computed using only $\tilde{\mu}_{a,\epsilon}^\ell$, is a high-probability upper bound on the true mean. Here, we introduce two specific indexes that satisfy this criterion.

For AdaP-UCB:
$$\mathrm{I}_a^\epsilon(t_\ell - 1, \alpha) = \tilde{\mu}_{a,\epsilon}^\ell + \sqrt{\frac{\alpha \log(t_\ell)}{2 \times \frac{1}{2} N_a(t_\ell - 1)}} + \frac{\alpha \log(t_\ell)}{\epsilon \times \frac{1}{2} N_a(t_\ell - 1)} \tag{7}$$

For AdaP-KLUCB:
$$\mathrm{I}_a^\epsilon(t_\ell - 1, \alpha) = \max\left\{q \in [0, 1] : d\left(\breve{\mu}_{a,\epsilon}^{\ell,\alpha}, q\right) \leq \frac{\alpha \log(t_\ell)}{\frac{1}{2} N_a(t_\ell - 1)}\right\}, \tag{8}$$

where $\breve{\mu}_{a,\epsilon}^{\ell,\alpha} = \mathrm{Clip}_{0,1}\left(\tilde{\mu}_{a,\epsilon}^\ell + \frac{\alpha \log(t_\ell)}{\epsilon \frac{1}{2} N_a(t_\ell - 1)}\right) = \min\{\max\{0, \tilde{\mu}_{a,\epsilon}^\ell + \frac{\alpha \log(t_\ell)}{\epsilon \times \frac{1}{2} N_a(t_\ell - 1)}\}, 1\}$ is the private empirical mean clipped between zero and one.

**Theorem 7.** *For rewards in $[0, 1]$,* AdaP-UCB *satisfies $\epsilon$-global DP, and for $\alpha > 3$, it yields a regret*
$$\mathrm{Reg}_T(\text{AdaP-UCB}, \nu) \leq \sum_{a:\Delta_a > 0} \left(\frac{16\alpha}{\min\{\Delta_a, \epsilon\}} \log(T) + \frac{3\alpha}{\alpha - 3}\right).$$

**Theorem 8.** *When the rewards are sampled from Bernoulli distributions,* AdaP-KLUCB *satisfies $\epsilon$-global DP, and for $\alpha > 3$ and constants $C_1(\alpha), C_2 > 0$, it yields a regret*
$$\mathrm{Reg}_T(\text{AdaP-KLUCB}, \nu) \leq \sum_{a:\Delta_a > 0} \left(\frac{C_1(\alpha)\Delta_a}{\min\{d_{\inf}(\mu_a, \mu^*), C_2\epsilon\Delta_a\}} \log(T) + \frac{3\alpha}{\alpha - 3}\right).$$

Both the upper bounds show that for low-privacy regime, the regrets of AdaP-UCB and AdaP-KLUCB are independent of $\epsilon$, and in high-privacy regime, they depend on the coupled effect. In Appendix E.4, we also derive problem-independent or minimax regret upper bounds for AdaP-UCB and AdaP-KLUCB, which are $\mathcal{O}(\sqrt{KT \log(T)} + \frac{K \log(T)}{\epsilon})$.

**Remark 1.** *$\alpha$ controls the width of the optimistic confidence bound. Specifically, it dictates that the real mean is smaller than the optimistic index with high probability, i.e. with probability $1 - \frac{1}{t^\alpha}$ at step $t$. The requirement that $\alpha > 3$ is due to our analysis of the algorithm. To be specific, the requirement that $\alpha > 3$ is needed to use a sum-integral inequality in the proof of Theorem 11. Since the dominant terms in the regret upper bounds of both* AdaP-UCB *and* AdaP-KLUCB *are multiplicative in $\alpha$, making $\alpha$ smaller will tighten the bound. We leave it for future work. We refer to Section G.3 for a detailed discussion on choosing $\alpha$.*

**Summary of Algorithm Design.** We propose three ingredients to design an $\epsilon$-global DP version of a bandit algorithm (Algo. 1). Firstly, add Laplace noise to make the empirical means private (Lemma 1). Secondly, compute the index of an arm using the private empirical mean of rewards collected from the last active episode of that arm and forget all the samples obtained before that. Thirdly, use adaptive episodes with doubling. At the beginning of an episode choose the arm with the highest index and play it double number of times than the length of the last active episode of the arm.

## 5 Experimental Analysis

In this section, we perform empirical evaluations to test two hypotheses: (i) AdaP-KLUCB is the most optimal algorithm among the existing bandit algorithms with $\epsilon$-global DP, and (ii) the

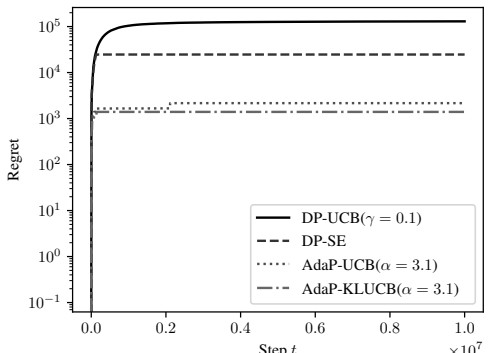
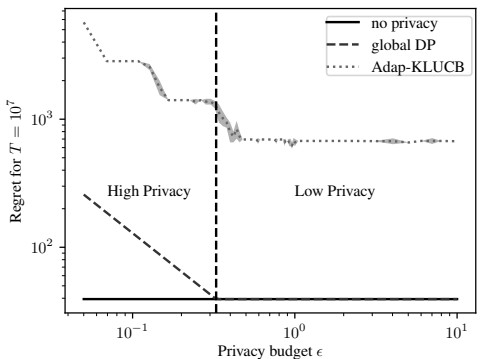

Figure 2: Evolution of regret over time for DP-UCB, DP-SE, AdaP-UCB, and AdaP-KLUCB with $\epsilon = 1$. Each algorithm is run 20 times with $T = 10^7$, and Bernoulli distributions with means $\{0.75, 0.625, 0.5, 0.375, 0.25\}$. AdaP-KLUCB achieves the lowest regret.

Figure 3: Dependence of lower bounds and regret of AdaP-KLUCB with respect to the privacy budget $\epsilon$. We run AdaP-KLUCB for 20 runs with $T = 10^7$. Echoing the theoretical analysis, regret of AdaP-KLUCB transits between privacy regimes and is independent of $\epsilon$ for low-privacy.

transition between high and low-privacy regimes is reflected in the empirical performance. Additional experimental results are deferred to Appendix G.

**Efficiency in Performance.** First, we compare performances of AdaP-UCB and AdaP-KLUCB with those of DP-SE and DP-UCB. We set $\alpha = 3.1$ to comply with the regret upper bounds of AdaP-UCB and AdaP-KLUCB. We assign $\gamma = 0.1$ for DP-UCB and $\beta = 1/T$ for DP-SE. We implement all the algorithms in Python (version 3.8) and on an 8 core 64-bits Intel i5@1.6 GHz CPU. We test the algorithms for Bernoulli bandits with 5-arms and means $\{0.75, 0.625, 0.5, 0.375, 0.25\}$ (as in Sajed and Sheffet (2019)). We run each algorithm 20 times for a horizon $T = 10^7$, and plot corresponding average and standard deviations of regrets in Figure 2. AdaP-KLUCB achieves the lowest regret followed by AdaP-UCB. Both of them achieve 10 times lower regret than the competing algorithms.

**Impact of Privacy Regimes.** In Figure 3, we plot regret of AdaP-KLUCB at $T = 10^7$ for a Bernoulli bandit with mean rewards $\{0.8, 0.1, 0.1, 0.1, 0.1\}$. We plot the average regret over 20 runs as a function of the privacy budget $\epsilon \in [0.05, 10]$. As indicated by the theoretical regret lower bounds and upper bounds, the experimental performance of AdaP-KLUCB demonstrates two regimes: a high-privacy regime (for $\epsilon < 0.3$), where the regret of AdaP-KLUCB depends on the privacy budget $\epsilon$, and a low privacy regime (for $\epsilon > 0.3$), where the regret of AdaP-KLUCB does not depend on $\epsilon$.

In brief, *our experimental results validate that* AdaP-KLUCB *is the most optimal algorithm for stochastic bandits that satisfies $\epsilon$-global DP, and performance of* AdaP-KLUCB *transits from high-to low-privacy regimes, where its performance turns independent of the privacy budget $\epsilon$.*

## 6 Conclusion and Future Works

We revisit bandits with $\epsilon$-global DP. We prove the minimax and problem-dependent regret lower bounds for stochastic and linear bandits indicating two regimes of hardness. We propose a framework to design an $\epsilon$-global DP version of an index-based optimistic bandit algorithm in a near-optimal fashion by utilising three ingredients: adding noise with Laplace Mechanism, using the empirical mean of rewards collected in the last active episode of an arm, and adapting episodes with doubling.

One limitation of our analysis that the lower and upper bounds defer by multiplicative constants. It would be a technical challenge to merge this gap. Another future direction is to derive regret lower bounds for other variants of DP, namely $(\epsilon, \delta)$-DP and Rényi-DP (Mironov, 2017), especially in the structured bandits compatible with Gaussian noise injection and also to extend the proposed proof techniques in these settings. For the algorithm design, it would be interesting to see how the proposed ingredients generalise to linear and contextual bandits (Shariff and Sheffet, 2018), and Markov Decision Processes (Vietri et al., 2020).

## Acknowledgments and Disclosure of Funding

This work is supported by the AI_PhD@Lille grant. D. Basu acknowledges Inria-Kyoto University Associate Team "RELIANT" for supporting the project. We also thank Philippe Preux for his support.

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
