# OpenReview forum: "When Privacy Meets Partial Information: A Refined Analysis of Differentially Private Bandits"
_NeurIPS.cc/2022/Conference — NeurIPS 2022 Accept_

### Official Review · Reviewer_75rR · 2022-07-06

**Rating:** 7
**Confidence:** 3
**Soundness:** 3 good
**Presentation:** 4 excellent
**Contribution:** 3 good

**Summary:**

This paper proves the minimax regret lower bound and problem-dependent regret lower bound for stochastic and linear bandits that quantify the hardness of bandits with $\epsilon$-global DP. The results reveal interesting phase transition phenomenon in that in the high-privacy regime, the hardness depends on a coupled effect of privacy and partial information about the reward distributions; while in the low-privacy regime, the regret of bandits with $\epsilon$-global DP reduces to that of bandits without privacy. In the algorithm, the authors then propose AdaP-UCB and AdaP-KLUCB methods which are $\epsilon$-global DP extensions of UCB and KL-UCB algorithms, respectively. AdaP-KLUCB is shown to be the first algorithm that both satisfies $\epsilon$-global DP and yields a regret upper bound that matches the problem-dependent lower bound.


**Questions:**

See above weakness part.

**Ethics Review Area:**

["I don’t know"]

**Strengths And Weaknesses:**

Strengths

1. In spite of a very theoretical paper, this paper is very well written and is easy to follow.

2. DP in bandit algorithms has received recent attentions. The hardness of local DP in bandits in terms of regret lower bound has been well studied in the literature. However, fundamental hardness of differentially private bandits with global DP is less studied, except for Shariff and Sheffet (2018).

3. The proposed algorithm is able to achieve a regret upper bound that matches the problem-dependent lower bound.


Weaknesses

1. Technical novelty beyond Shariff and Sheffet (2018). Shariff and Sheffet (2018) studied the problem-dependent lower bound on regret for stochastic bandits of Bernoulli reward with $\epsilon$-globally DP. This paper extends the results to general reward case. It is important to explicitly highlight the technical contributions beyond Shariff and Sheffet (2018) .

2. In the experiments, the proposed algorithms were compared to the benchmark methods DP-SE and DP-UCB. However, there are some recent developments, e.g., DP-TS by Hu and Hegde (2022), on differentially private stochastic bandits. DP-TS and its lazy version are shown to be superior over DP-SE. It is important to compare with the state-of-the-art methods.

Hu and Hegde (2022), Near-Optimal Thompson Sampling-based Algorithms for Differentially Private Stochastic Bandits, UAI.

3. Can authors provide some discussion on the requirement $\alpha > 3$ in algorithms and theorems? I noticed that $\alpha =1$ actually works the best as shown in the additional simulation results in the supplementary materials.

4. The authors claim that Figure 3 show that in the low privacy regime ($\epsilon > 0.3$), the regret of AdaP-KLUCB does not depend on $\epsilon$. However, what I observe is that when $\epsilon > 0.3$, the regret first decreases when $\epsilon$ increases and then stabilizes. This step-shape decreasing pattern is consistent even when $\epsilon < 0.3$. Therefore, more convincing numerical evidence would be helpful to justify this claim.

---

> ### Author Response · Authors · 2022-08-02
> **Author response to specific questions**
>
> We thank the reviewer for spending valuable time and for the constructive feedback.
>
> **Technical contributions in lower bounds beyond [1]:** Our technical contributions beyond [1] consist of: (a) extending the Karwa-Vadhan lemma (Lemma 6.1, [13]) to a sequential setting (Lemma 2), (b) presenting a novel sequential information processing lemma under $\epsilon$-global DP that controls the difference between the outcome streams of a differentially private policy when interacting with two different bandit instances (Theorem 10), and (c) proposing a generic proof structure leading to new lower bounds for different bandit settings. These new technical tools, such as the sequential Karwa-Vadhan lemma or the proofing technique are generalisable to structured bandits, and have wider applications as pointed out by Reviewer 8hyR. We mentioned this in Technical Tools (line 78) and Related Work (line 88). We will clarify and emphasize these Technical Tools in the updated draft.
>
> **Reference to DP-TS:** We thank the reviewer for pointing us to Hu and Hegde (2022). It is indeed a good idea to compare our algorithms to DP-TS as TS methods are effective in practice.
> However, given the timeframe (the article appeared on OpenReview in April) and the fact that it's still unpublished work (to appear in UAI 2022), it was not possible for us to be aware of this reference while doing our work. Here, we provide a statement that we would like to add in the Related Work section while referring to Hu and Hegde (2022):
>
> "DP-TS [Hu and Hegde, 2022] aims to achieve global DP with a Thompson sampling-based approach. AdaP-KLUCB achieves our problem-dependent lower bound with Kullback-Leibler (KL) indistinguishability gap, while DP-TS cannot. Even in non-private bandits, TS is not known to achieve the Kullback-Leibler (KL) indistinguishability gap while KL-UCB is known to achieve it. Thus, From a theoretical point of view, AdaP-KLUCB incurs lower regret than DP-TS."
>
> **Discussion on $\alpha > 3$:** We refer to the general comment section for a detailed discussion on choosing $\alpha$, its implications, and possible future works.
>
> **Step-shape decreasing pattern in Figure 3:** We would like to thank the reviewer for this wonderful insight. As the step-shape decreasing pattern in Figure 3 is indeed present even in the low privacy regime ($\epsilon < 0.3$), we agree that stopping our experiments at $\epsilon = 1$ might lead to confusion regarding the evidence. We re-run the experiment with the x-axis ($\epsilon$) expended till $\epsilon = 20$ and our claim is still validated as the regret remains constant with respect to the x-axis (i.e. $\epsilon$). We will include the new figure in the final version.

---

> > ### Comment · Reviewer_75rR · 2022-08-08
> > **Thanks for the response**
> >
> > Thanks for the response. My major concerns have been nicely addressed.

---

### Official Review · Reviewer_B8Us · 2022-07-09

**Rating:** 5
**Confidence:** 5
**Soundness:** 2 fair
**Presentation:** 2 fair
**Contribution:** 2 fair

**Summary:**

This paper considers DP bandits under the central model. The authors present both minimax and problem-dependent lower bounds for both MAB and linear bandits. Then, they also establish a general method for designing private bandit algorithms based on non-private ones. Extensive numerical simulations are conducted to validate the theoretical results.

**Questions:**

- My first concern is about the proof of the privacy guarantee in Section 4. The tricky part is that the adaptive batch schedule also depends on private data (rewards). Thus, an adversary can observe the batch schedule to infer the private data. More specifically, changing one reward in the sequence will also change the follow-up batch schedule. The standard proof cannot be directly applied here. I believe this can be handled by a careful but non-trivial argument, but I will first leave it to the authors in the rebuttal period.

- My second concern is more general. This relates to the commonly-used notion of DP in bandits. In particular, the neighboring relation is about two neighboring reward sequences, i.e., differ in the reward value at only one particular time slot. This is inspired by the standard continual model of DP. However, this is not the right notion for DP in MAB. To see it, this notion requires that if one changes a reward at time $t$, the reward obtained at $t+1, t+2,...$ are still the same by the requirement of the neighboring definition. This is obviously not the case in MAB, since the policy would be different at   $t+1, t+2,...$ due to the change of reward at time $t$, and hence the agent will pull different arms and hence different rewards will be observed. Note that this is in sharp contrast to the classic continual observation DP model where the typical task is simply to count the current counts in the online stream data, where there is no correlation between the data across time slots, and hence the standard neighboring relation makes sense in terms of privacy protection. I think one can also propose a more meaningful notion of DP in bandits (with only a minor change of the current proof), but again I will leave it to the authors in the rebuttal period.

- In fact, the intuition for the above two questions are quite related.

**Limitations:**

Yes,  limitations and potential negative societal impacts have been addressed.

**Strengths And Weaknesses:**

**strengths**
- A comprehensive study of lower bounds under the central model

**Weaknesses**
- The proof for the privacy guarantees in the upper bound part seems to need more care

---

> ### Author Response · Authors · 2022-08-02
> **Author response to specific questions**
>
> We would like to thank you for your time, thorough feedback, and the kind words about the comprehensiveness of our study.
>
> **Meaningfulness of $\epsilon$-global DP to study privacy in bandits:** In order to clarify our setting, we provide an example.
>
> Let us consider that we want to compare the effectiveness of $k$ medicines (corresponding to $k$ arms) by testing them on $T$ patients. At each step $t \in [T]$, we choose a medicine $a_t \in [k]$. We test it on a new patient $\texttt{patient}_t$ and obtain a reward $r_t$, i.e. whether the patient is cured (say $1$) or not ($0$). From a bandit algorithm perspective, the goal is to maximise the cumulative reward, i.e. the number of patients getting cured. From the privacy point of view, the goal is to construct a randomized policy, that in a parallel universe with a different patient at some step $t \in [T]$, still recommends the same sequence of medicines.
> In this case, the output of the bandit algorithm is the choices of medicines and its input is the reactions (cured/not cured) of the $T$ patients to the corresponding medicines. In this setting, the definition of neighbouring inputs and global DP used in our paper and also in [18,25,20,3,12] makes sense.
>
>
> From a high-level perspective, the confusion as mentioned in the review arises if the rewards are perceived as random variables. In that case, rewards are coupled with actions under a stochastic assumption. In contrast, in the $\epsilon$-global DP definition, the sequence of rewards is a fixed "instantiation" of these random variables. A randomised policy takes this instantiation of reward sequence as an input to produce a sequence of actions. Specifically, Sec 2.1. of [18] addresses your concern by pointing out that "we cannot use any stochastic assumption on the reward functions for privacy guarantee". We refer to Section A of the appendix where we rigorously define the bandit policy (the function to make DP), its input (a sequence of instantiated rewards) and its output (a sequence of actions). With this definition, the neighbouring relation among inputs as in our paper and previous works stays valid.
> We also refer to [3] for details that compares different possible definitions of DP for bandits and unifies them under global (our work) and local DP.
>
> **Proof of the privacy guarantee:** We would like to thank the reviewer for such a detailed remark. We briefly explain the validity of our proof here.
>
> It is indeed a non-trivial argument to handle the batch schedule and the intuition is related to the first concern as pointed out by the reviewer. To prove the privacy of the algorithm, we fix two neighbouring sequence of rewards $r^T = \{ r_1, \dots, r_T \}$ and $r'^T = \{r'_1, \dots, r'_T\}$. We also fix a sequence of actions $a^T = \{a_1, \dots, a_T\}$.
> The goal would be to prove that: $Pr(\pi(r^T) = a^T) \leq e^\epsilon Pr(\pi(r'^T) = a^T)$.
> The episode schedule, i.e $t_1 < t_2 \dots <t_\ell$ could be directly inferred from the fixed sequence of actions $a^T$ which is indeed public information. Then, we use Lemma 1 with this fixed schedule and post-processing property of DP to conclude the privacy proof. We will add this comment in the privacy proof section of the updated draft.

---

> > ### Comment · Reviewer_B8Us · 2022-08-06
> > **Response to authors**
> >
> > Thanks for the response.
> >
> > In my opinion, a more proper definition for DP bandit should consider the change of one user (note that not the standard user-level DP). For example, in the DP tabular MDP paper [R1] (which is a strict generalization of MAB) considers this. A more older but classic paper on DP (adversary) bandit is [R2], where each user (say patient $t$) is represented by a function $f_t$ (note that stochastic bandit can be viewed as a special case of adversary bandit).
> >
> > Regarding the proof for the privacy guarantee, I think one needs to formally use the adaptive composition (i.e., chain rule of conditional probability). For example, see the proof of Theorem 3 in [R2] (in particular the subtlety pointed out by the authors). I think this chain rule is required even one considers the current DP definition.
> >
> > Can the authors comment on the above discussions?
> >
> > A follow-up question. After a closer look at the current minimax lower bound for MAB established in this paper, it seems to me that it is looser by factor $\log T$ than the one used in [20] (see remark after Theorem 4.4) and the one established for the tabular MDP (which reduces to MAB by setting $H=1$ and $S=1$) in [R1].
> >
> >
> > Can the authors comment on the above observation?
> >
> > References:
> >
> > [R1] Vietri, G., Balle, B., Krishnamurthy, A. and Wu, S., 2020, November. Private reinforcement learning with pac and regret guarantees. In International Conference on Machine Learning (pp. 9754-9764). PMLR.
> >
> > [R2] Guha Thakurta, A. and Smith, A., 2013. (Nearly) optimal algorithms for private online learning in full-information and bandit settings. Advances in Neural Information Processing Systems, 26.

---

> > > ### Author Response · Authors · 2022-08-07
> > > **Author response to additional questions**
> > >
> > > We thank the reviewer for the interest in our response and the corresponding follow-up questions.
> > >
> > > **More details on the meaningfulness of $\epsilon$-global DP.** We thank the reviewer for mentioning the references [R1] and [R2] as our work uses the same setting of DP in both these papers. $\epsilon$-global DP is the exact notion used by [R2] in the section **Private Online Learning: Bandit Setting**, by taking $\omega_t = a_t$ and $f_t(\omega_t) = - r_t$ in their definition (since $f_t$ are loss functions there).
> > >
> > > Also, the JDP notion used by [R1] for MDPs reduces to $\epsilon$-global DP when $H=1$ and $S=1$ (as pointed out by the reviewer himself in the follow-up question when asking to compare our minimax results to the ones in [R1]). The "Episodic RL Protocol" (Algorithm 1 in [R1]) captures perfectly the example of medicines and patients given above, when rather than interacting with the same user for a whole trajectory in RL, you only get one reward in MAB ($H=1$). In addition, the lower bounds proved in [R1] rely on a reduction of JDP in MDPs to $n=S-2$ MAB instances satisfying $\epsilon$-global DP (Section C.4 in [R1]) and then using the lower bounds of [1], which are lower bounds for $\epsilon$-global DP bandits. This proves that the notions of DP used in [R1], [R2] and our work are all the same.
> > >
> > > **The chain rule in the proof of the privacy guarantee.** We thank the reviewer for this precise remark. We implicitly use the exact intuition as in the proof of Theorem 3 in [R2] in our privacy proof: the tricky part (as pointed out by [R2]) is that the private means at a step $t$ depend adaptively on the **publicly** released actions before step $t$, but since those actions are already private, by "adaptive" post-processing, you can deal with them inside the algorithm as if they were fixed public data that do not leak any information.
> > >
> > > To avoid confusion, we make our argument more formal by stating the exact calculations and chain rules used for proving the privacy.
> > >
> > > *Claim:* Fix two neighboring reward streams $r^T =\{r_1, \dots, r_T\}$ and $r'^T = \{r'_1, \dots, r'_T\}$. This implies that $\exists j \in [1, T]$ such that $r_j \neq r_j'$ and $\forall t \neq j$, $r_t = r_t'$. We also fix a sequence of actions $a^T = \{a_1, \dots, a_T\}$. We want to show that:
> > > $Pr(\pi(r^T) = a^T) \leq e^\epsilon Pr(\pi(r'^T) = a^T)$.
> > >
> > > *Proof sketch:*
> > >
> > > - Since $r^{j - 1} = r'^{j - 1}$, $Pr(\pi(r^{j - 1}) = a^{j - 1}) = Pr(\pi(r'^{j - 1}) = a^{j - 1})$.
> > > - Let $t_\ell \leq j < t_{\ell + 1}$ and $t_{\ell'} \leq j < t_{\ell'+ 1}$ be the episodes corresponding to $r^T$ and $r'^T$ resp. Since $r^{j - 1} = r'^{j - 1}$, we get that $\ell = \ell'$. Thus, $Pr(\pi(r^{t_{\ell + 1}}) = a^{t_\ell  + 1}) = Pr(\pi(r'^{t_\ell + 1}) = a^{t_\ell + 1})$.
> > > - Let $\tilde{\mu}\_{a_j,\epsilon}^{\ell}$ and $\tilde{\mu}\_{a,\epsilon}^{' \ell}$ be the private means of arm $a_j$ computed in the episode $[t_\ell, t_{\ell + 1}]$, by the Laplace mechanism, for every interval $I \in \mathcal{R}$, $Pr(\tilde{\mu}\_{a_j,\epsilon}^{\ell} \in I) \leq e^\epsilon Pr(\tilde{\mu}\_{a_j,\epsilon}^{'\ell} \in I)$.
> > > - Finally, since $\{r_{j+1}, \dots, r_T \} = \{r'_{j+1}, \dots, r'_T \}$, $Pr(\pi(r^T) = a^T |  \tilde{\mu}\_{a_j,\epsilon}^{\ell} \in I) = Pr( \pi(r'^T) = a^T | \tilde{\mu}\_{a_j,\epsilon}^{'\ell} \in I)$
> > >
> > > Now, we conclude the argument by using a chain rule. QED.
> > >
> > > We refer to the proof of Theorem 4.1 (DP-SE) in [20] that explains the intuition behind the proof presented here.
> > >
> > > **Comparaison of the minimax lower bound to [20] and [R1].** Both minimax lower bounds in [20] and [R1] are for Bernoulli distributions since they rely on the lower bound of [1], which is specific to Bernoulli distributions. However, our Theorem 2 is a minimax lower bound for Gaussian distributions (line 639 in the appendix explains the class of distributions used, we will add a line defining it in the main paper too), which is a new result, independent of [1].

---

> > > > ### Comment · Reviewer_B8Us · 2022-08-08
> > > > **Response to authors**
> > > >
> > > > Thanks for the timely response.
> > > >
> > > > >  $\epsilon$-global DP is the exact notion used by [R2] in the section Private Online Learning: Bandit Setting by taking $w_t = a_t$ and $f_t(w_t) = -r_t$.
> > > >
> > > > They are **NOT** the same based on my understanding. As the authors already note that here $r_t = - f_t(w_t)$ (the sign does not matter for this discussion). In [R2], the neighboring relation of DP is on $f_t$ (see their Eq. 2), i.e., the function itself while in your current paper, the neighboring notion is on $r_t$, which is just one particular value. The one in [R2] is a proper one, which can be used to capture real-life applications, i.e., to capture the person rather than a single feedback.
> > > >
> > > > >  This proves that the notions of DP used in [R1], [R2] and our work are all the same.
> > > >
> > > > I don't think so. Even let $H=1$ and $S=1$, the one in [R1] is not the same as yours. In [R1], they directly consider a neighboring relation on users, and more importantly, each user is identified by a function -- that is, the reward response the user can give to all the possible actions (see the third line after Algorithm 1 in http://proceedings.mlr.press/v119/vietri20a/vietri20a.pdf).
> > > >
> > > > > minimax lower bound
> > > >
> > > > Let me clarify my question a little bit. The current minimax lower bound in [20, R1] is $\Omega(\sqrt{KT} + \frac{K\log T}{\epsilon})$ (for the class of Bernoulli distributions). The current paper establishes $\Omega(\sqrt{KT} + \frac{K}{\epsilon})$ for Gaussian. **Is the difference of $\log T$ in the privacy term due to different class of distributions?** In other words, if one considers the Bernoulli distribution, can the current proof technique yield the same minimax lower bound as in [20, R1]? I tend to think it may not be the case.

---

> > > > > ### Author Response · Authors · 2022-08-09
> > > > > **Author response to additional questions**
> > > > >
> > > > > We thank the reviewer for further queries and feedback.
> > > > >
> > > > > **Neighboring on $r_t$ vs on $f_t$**. The main concern is that the notion of neighboring used in our work relies on $r_t$ (a particular value of the **received** reward), while a stronger notion of neighboring would consider the whole vector $f_t$ (all potential answers of one user). Lemma 1 of [3] shows that, in a bandit setting, these two notions are in fact equivalent. Intuitively, if a bandit algorithm is private with respect to all changes in the received feedback, for all possible sequences of actions, it is private for all the potential answers too. This validates our claim that [R1], [R2] and our work use the same notion of DP. We refer to [3] as it discusses in depth all possible definitions of DP for bandits and unifies them under global (our work) and local DP.
> > > > >
> > > > > **More details on the comparison of the minimax lower bounds to [20] and [R1]**. We thank the reviewer for the clarification. Indeed, our minimax lower bound is for Gaussian distributions and adapting the proof to Bernoulli distributions seems to not recover the extra $\log(T)$ factor. A potential reason is the fact that both [20] and [R1] rely on the result of [1], which has a strong assumption that the policy is a consistent one. Our result is a more "worst-case" lower bound, since it is a minimax lower bound on the whole class of $\epsilon$-global DP policies (not only consistent ones). The standard minimax lower bounds and corresponding proofs in non private bandits [Theorem 15.2, Lattimore and Szepsvari, Bandit algorithms, 2020] follow the similar setting like ours rather than the added consistency assumption of [1, 20, R1].

---

> > > > > > ### Comment · Reviewer_B8Us · 2022-08-09
> > > > > > **Response to authors**
> > > > > >
> > > > > > Thanks for the authors' update.
> > > > > >
> > > > > > > Neighboring relation...
> > > > > >
> > > > > > Thanks for pointing out Lemma 1 of [3], which seems to resolve my concern. Again, the key idea is nothing but chain rule of conditional probability:)
> > > > > >
> > > > > > > Minimax lower bound...
> > > > > >
> > > > > > Based on my understanding of the authors' response, it seems that the lower bound in the current work [20, R1] is not **minimax**, am I correct? Do the authors in [20, R1] mess up with the problem-dependent bound with the minimax bound? In particular, to my best knowledge, the consistency assumption is often only used to derive problem-dependent lower bounds. (BTW, can the authors give an exact pointer of the result in [1] that has been used in [20, R1]?)
> > > > > >
> > > > > > BTW, in your reference [3], there also exists a result for minimax lower bound under global DP, which is very different from the current one. It seems that there is no comparison with the result in [3] and can the authors comment on this？

---

> > > > > > > ### Author Response · Authors · 2022-08-09
> > > > > > > **Author response to additional questions**
> > > > > > >
> > > > > > > We are glad that some of your concerns are clarified and thank you for further feedback.
> > > > > > >
> > > > > > > **More details on the minimax lower bounds**. The exact result from [1] used by [20, R1] is the Corollary 15, which uses Claim 14, that has a "consistency assumption". Indeed, this is a problem dependent lower bound, improperly used by [20, R1] in their minimax lower bounds (as they are applying it without mentioning any consistency assumption). This is one of the reasons that motivates our work to refine and explicit the lower bounds in private bandits literature, in a resonance to the existing results for non-private bandits [ref. Lattimore and Szepsvari, Bandit algorithms, 2020].
> > > > > > >
> > > > > > > [3] uses the same structure of proof as our work. However, their KL-divergence decomposition in Lemma 6 is not tight compared to ours, which makes our lower bound tighter than theirs. Also, [3] needs to assume that the reward distributions are Lipschitz continuous and the corresponding Lipschitz constant appears in their bound. We do not need such assumption.
> > > > > > >
> > > > > > > Additionally, we believe that our minimax lower bound is achievable as we have an algorithmic sketch that achieves this lower bound and we are working on to improve it.
> > > > > > >
> > > > > > > We thank you again for your careful review and following interactions. We hope to have answered your concerns and we pledge to include relevant discussions in the revised version. If you feel that your concerns have been answered, we would appreciate it if you can adjust your score accordingly.

---

> > > > > > > > ### Comment · Reviewer_B8Us · 2022-08-09
> > > > > > > > **Response to authors**
> > > > > > > >
> > > > > > > > Thanks for the update.
> > > > > > > >
> > > > > > > > I hope the authors can add one paragraph to carefully discuss the current situation of minimax lower bounds for private bandits so as to better sell the established result in the paper. This is important since it seems to me that there exists some misunderstanding in [20, R1] based on the authors' explanations.
> > > > > > > >
> > > > > > > > I have updated my score accordingly.

---

> > > > > > > > > ### Author Response · Authors · 2022-08-09
> > > > > > > > > **Author response**
> > > > > > > > >
> > > > > > > > > We are glad that all of your concerns have been addressed and thank you for raising your score. We will add a paragraph explaining the details discussed here and the comparaison to [1, 20, R1, 3] after Theorem 2.

---

### Official Review · Reviewer_8hyR · 2022-07-10

**Rating:** 8
**Confidence:** 4
**Soundness:** 3 good
**Presentation:** 4 excellent
**Contribution:** 4 excellent

**Summary:**

This paper studies the regret minimization problem for stochastic (Bernoulli and linear) bandits under the notion of global differentially privacy. This is a thorough study of the problem where the authors propose lower bounds quantifying fundamental problem hardness, an algorithmic framework and upper bound on the regret for the proposed algorithms under this framework. These results are supported by sound numerical findings. Some highlights:

-- Two sets of lower bound results are provided: problem-dependent and minimax. The key lemma driving these results is a non-trivial extension of Karwa-Vadhan lemma which bounds the differential privacy when the data is generated from non-identical distributions.

-- The algorithmic developments are novel from a bandit perspective more-so than from a privacy perspective, particularly, the idea of adaptive phases which is the key driver in ensuring the privacy of the algorithm.


**Questions:**

-- A natural idea would be to use a tree-based aggregation technique for adaptively keeping track of the means. It is not completely clear to me how adaptively using doubling horizon ensures the same privacy level. Further, the privacy proofs do not require composition which is surprising. I would like to be convinced if this is due to the definition of Global DP or a consequence of the algorithmic template.


**Limitations:**

A contextual bandit extension would be nice esp. since the authors get rid of tree-based aggregation. But there the adaptive-doubling idea might not work since the minimum eigenvalue of the covariance matrix would need to be updated regularly in order to have consistent parameter estimate.

**Strengths And Weaknesses:**

Strengths:

-- Non-trivial novel theoretical contributions for both lower bound and upper regret bound. Some of the key results (for eg., Theorem 10) could be of independent interest and have wider application. The key implication of the lower bound is the phase transition behavior of bandits depending on the privacy budget. This has implications particularly because in the low-privacy regime, the problem is equivalent to the non-private setting.

-- The general purpose framework proposed for algorithmic design unifies existing bandit algorithms. This is a much deeper contribution than proposing and analyzing a single algorithm. I believe this makes the contribution very fundamental.

-- Numerical results are also done for a variety of privacy budget though it would interesting if they could demonstrate the lower bounds as well as study the dependence on $\alpha$. A

Weakness:

-- More discussion regarding the parameter $alpha$ and how to tune it. A small discussion regarding this possibly in the main body of the paper.

---

> ### Author Response · Authors · 2022-08-02
> **Author response to specific questions**
>
> We would like to thank the reviewer for the time spent reviewing, careful reading, and kind words about the novelty and the significance of the contributions. It is really encouraging for us.
>
> **The tree-based algorithm and doubling the episode:** As suggested by the reviewer, it is indeed our algorithmic framework that allows us to achieve $\epsilon$-global DP without the need for either composition or the tree mechanism. Specifically, having episodes with forgetting (i.e each mean only uses samples from the last active episode), allows us to achieve the same privacy guarantee as the tree mechanism ($\epsilon$-DP) while adding less noise. The main difference is that the tree mechanism keeps track of all the history (since its original use is for counting queries, where all the history is important) and hence has larger sensitivity to a change in the rewards. In fact, DP-UCB [18, 25] leverages a tree-based mechanism but has an extra multiplicative $\log(T)^{1.5}$ compared to the regret lower bound. Our goal in this work was to get rid of the extra factor. The main intuition behind our algorithmic framework is that for bandits, it's possible to only access a part of the history (the last active episode of each arm) and still achieve low regret. We formalize this intuition in Lemma 1.
>
> **Comments on $\alpha$:** We refer to the general comment for a detailed discussion on choosing $\alpha$, its implications, and possible avenues for improvement.
>
> **Limitation: Extension to contextual bandits:** As you have aptly pointed out, extending our algorithmic framework to contextual bandits is an interesting future work that we have mentioned in the paper and are currently investigating. You are right that adapting the doubling episode scheme to contextual and structured bandits might be trickier as we have to be careful of the corresponding structure.

---

> > ### Comment · Reviewer_8hyR · 2022-08-02
> > **Additional questions**
> >
> > --
> > Can you elaborate a bit more on this idea of adaptive doubling and why it works intuitively? In particular, how are you keeping track of remote past information that was not a part of last episode? I would think that there is information leakage possible here.
> >
> > --
> > How is $\varpesilon$-DP different from joint differential privacy in the streaming setting?

---

> > > ### Author Response · Authors · 2022-08-03
> > > **Author response to additional questions**
> > >
> > > We thank the reviewer for the interest in our response and the corresponding follow-up questions.
> > >
> > > **Details of the adaptive episodes and privacy implications**
> > >
> > > To clarify our algorithmic framework, let us consider the illustration of Example 2. In particular, let us move to step $t_4 = 7$. To compute the indexes of the two arms, the policy only uses the privatized means: $\frac{r_4 + r_5}{2} + Lap(\frac{1}{2 \epsilon})$ for arm 1 and $r_6 + Lap(\frac{1}{\epsilon})$ for arm 2. In particular, the values of $r_1$, $r_2$ and $r_3$ will never be used again by the policy, to make any decision, at any step $t > t_4$. Intuitively, not using an individual's input is the highest privacy an algorithm could guarantee. Our framework achieves that thanks to the forgetting in the episodic scheme (one can find a similar trick in DP-SE [20]). In contrast, the tree mechanism in DP-UCB [18, 25] uses all the sequences of rewards in the history to compute its indexes. The value of $r_1$, for example, is affecting all the actions chosen by the algorithm, hence the sensitivity of the algorithm to a change in $r_1$ is very high.
> > >
> > > For your second concern, all the rewards which are not part of the last episode are totally discarded (i.e. not even stored). The only statistics kept and used by the algorithm are the **privatized** means computed using the reward collected in the last active episode (e.g. the privatized mean for arm 1 at step $t_4=7$ is $\frac{r_4 + r_5}{2} + Lap(\frac{1}{2 \epsilon})$). Since these statistics are already private (due to the addition of the corresponding Laplace noise), all the decisions made using them are private too by the post-processing property of DP. Consequently, there is no information leakage there.
> > >
> > >
> > > **Difference between $\epsilon$-global DP and Joint Differential Privacy**
> > >
> > > Joint Differential Privacy [1] is a formalization of DP for contextual bandits, where the source of the sensitive information is both in the rewards and the contexts. This is not the case for stochastic and linear bandits, where the rewards are the only sensitive data. However, since stochastic bandits can be seen as a special case of contextual bandits with one unique context, the two definitions coincide in this case.

---

### Official Review · Reviewer_9pTm · 2022-07-20

**Rating:** 6
**Confidence:** 3
**Soundness:** 3 good
**Presentation:** 3 good
**Contribution:** 3 good

**Summary:**

This work investigates $\epsilon$-global differential privacy for multi-armed bandits. For both stochastic and linear bandits, it derives refined instance-dependent and global lower bounds on the regret for differentially private learning, in that they quantify the burden of privacy in two distinct regimes. Crucially, in the low-privacy regime, the privacy constraint doesn't make the bandits problem any harder. Next, the work formulates a general $\epsilon$-global strategy for stochastic bandits and shows that two algorithms based on it have regret upper bounds with privacy regime patterns like those in the lower bounds. This is confirmed empirically, where the two privacy regimes are evident.

**Questions:**

1. It would be useful to clarify either in the abstract or in "Our Contributions" that the regret lower bounds effectively generalize the claims of [1] in generalizing it beyond Bernoulli to general distributions. My point is that there was prior work hinting at distinct regimes of privacy and I think it's a good idea to acknowledge that at the outset.
2. It would be useful to discuss the gap between the lower bounds and the upper bounds for AdaP-UCB and AdaP-KLUCB. If feasible, instance-independent upper bounds would also be useful to have.
3. In Algorithm 1 (line 253), please clarify where $I_a^\epsilon$ comes from. You mention index-based algorithm but never actually define $I_a^\epsilon$.
4. Please fix some minor typos (e.g. "Algortihm") through the manuscript.

**Limitations:**

Not applicable.

**Strengths And Weaknesses:**

Strengths:
- The paper touches contributes to a very important theme in the context of differential privacy for bandits - the cost of obtaining this privacy - and identifies two distinct regimes of privacy preservation. Empirical results confirm the theory.
- It materializes a simple $\epsilon$-global DP strategy applicable to existing index-based bandit algorithms. This strategy clearly incorporates features that are known/desirable in the context of DP algorithms (e.g. Laplacian noise, private empirical means using little data).

Weaknesses: Please see questions.

---

> ### Author Response · Authors · 2022-08-02
> **Author response to specific questions**
>
> We would like to thank the reviewer for the time spent reviewing and for the precise remarks.
>
> **Generalisation of the existing lower bound for Bernoulli reward:** Our problem-dependent regret lower bound for stochastic bandits is indeed a generalization of [1] beyond Bernoulli reward distributions. We have mentioned that in Related Work (line 82), after stating Theorem 3 (line 200) and also as a remark in the appendix after the proof (line 689). We will also mention this in 'Our Contributions'.
>
> But our lower bound (Theorem 3) also provides a novel observation that the difficulty of a bandit problem with DP in the high-privacy regime depends both on the TV-indistinguishability gap ($t_{inf}$) and $\epsilon$. This interaction between privacy and partial information is not present in [1], which is a recurring observation of our lower bounds.
> We refer to the general comment for further details regarding the novelty and impact of our lower bounds and proof techniques.
>
> **Gap between lower and upper bounds:** We thank the reviewer for these two suggestions that would make a great addition to our work.
> We will add a comment explaining the difference in the "multiplicative constant" between the regret upper bounds of Adap-UCB and Adap-KLUCB compared to the lower bounds. We will mention it after stating the results of Theorems 7 and 8.
>
> **Instance-independent regret upper bounds:** Instance-independent or minimax regret upper bounds could be also provided for Adap-UCB and Adap-KLUCB and we will add them too. They are of order $O(\sqrt{K T \log(T) )} + \frac{K \log(T)}{\epsilon})$.
>
> **Formatting: $I^\epsilon_a$ and typo:** Thanks for pointing out these issues. We will introduce $I^\epsilon_a$ in the comments used inside Algorithm 1 and refer to the corresponding equations. We will also proofread the paper to eliminate the typos.

---

### Author Response · Authors · 2022-08-02
**General Comment**

We would like to thank the reviewers for acknowledging the strengths and soundness of the contribution as well as for their thoughtful comments and efforts towards improving the manuscript. In the following, we highlight general concerns of reviewers that were common and our effort to address these concerns. We then address comments specific to each reviewer by responding to them directly.

As pointed out in the reviews, our main goal is to propose a thorough examination of the well-studied problem of bandits with $\epsilon$-global DP. We provide a generic proof technique that we use to generate four regret lower bounds (minimax and problem-dependent, for stochastic and linear bandits) that all reflect a phase transition behaviour depending on the privacy budget $\epsilon$. We also propose an algorithmic recipe to make any index-based bandit algorithm $\epsilon$-global DP, that we use to instantiate Adap-UCB and Adap-KLUCB and show that they achieve the problem-dependent regret lower bound. As also pointed out in the reviews, some of the theoretical contributions (extension of Karwa and Vadhan Lemma 2, sequential information processing lemma Theorem 10 and generic regret analysis of bandit algorithms with adaptive episodes Theorem 11) could be of general use beyond our work.

Now, we would like to address two recurring comments:

- **Lower bounds beyond generalizing [1]**: Our problem-dependent regret lower-bound for stochastic bandits indeed generalizes the lower bound of [1] beyond Bernoulli distributions. We have mentioned that in Related Work (line 82), after stating Theorem 3 (line 200) and also as a remark in the appendix after the formal proof (line 689). We will also mention this in 'Our Contributions' as suggested by Reviewer 9pTm.

    But our generic proof techniques to generate the lower bounds, the corresponding extension of the Karwa-Vadhan lemma, and cementing the fact that global DP bandit algorithms can achieve as low regret as non-private bandit algorithms in the low-privacy regime in different settings are 'non-trivial', 'novel', and can aid in 'wider applications' as reviewers kindly pointed out.

    For instance, our generic proof technique lead to three other 'new' lower bounds for stochastic bandits (minimax, Theorem 2) and linear bandits (minimax, Theorem 4  and problem-dependent, Theorem 5). These are novel bounds independent of the result in [1] and important to spearhead near-optimal global-DP bandit algorithms in these settings.

    Additionally, our problem-dependent lower bound for stochastic bandits (Theorem 3) provides a novel observation that the difficulty of a bandit problem with global DP depends on the TV-indistinguishability gap ($t_{inf}$). This was not known in [1]. Rather, the regret lower bound due to privacy in [1] ($\frac{K \log(T)}{\epsilon}$) seems to be independent of the hardness of the bandit instance. Our result fills up this missing link between the interaction of privacy and partial information.

- **Comments on $\alpha$**: $\alpha$ controls the width of the optimistic confidence bound. Specifically, it dictates that the real mean is smaller than the optimistic index with high probability i.e with probability $ 1 - \frac{1}{t^\alpha}$ at step $t$.
    The requirement that $\alpha > 3$ is purely due to our analysis of the algorithm. This also happened in the classic bandit literature. [Bubeck and Cesa-Bianchi, 2012] required $\alpha > 2$ for analyzing UCB-type algorithms. This condition was dropped for UCB after a more involved and improved technical analysis ("On Upper-Confidence Bound Policies for Non-Stationary Bandit Problems" Garivier and Moulines). But our main goal was to provide an algorithmic framework and analysis that can render any index-based bandit algorithm into a global DP one. Thus, we restrained from the specific and involved analysis.

    Since the dominant term in the regret upper bound of both Adap-UCB and Adap-KLUCB is multiplicative in $\alpha$, $\alpha=1$ works better in practice, as shown in Figure 8. To be more specific, the requirement that $\alpha > 3$ is needed to use a sum-integral inequality to bound Term 2 in Step 3, line 850. We leave it for future work to relax this requirement. We will add this discussion in the final draft.

[Bubeck and Cesa-Bianchi, 2012] Sébastien Bubeck, and Nicolo Cesa-Bianchi. "Regret analysis of stochastic and nonstochastic multi-armed bandit problems." Foundations and Trends® in Machine Learning 5.1 (2012): 1-122.

---

### Meta-Review · Area_Chair_Q2qh · 2022-08-29

**Recommendation:** Accept
**Confidence:** Certain

**Metareview:**

This paper studies the problem of multi-armed bandits under differential privacy. The reviewers are all positive about the results and presentation of the paper.

**Award:**

No

---

### Decision · Program_Chairs · 2022-09-14

Accept